# Thailand's Sex Entertainment: Alienated Labor and the Construction of Intimacy

**Petra Lemberger** [1],* and **Tony Waters** [2,3],*

1 Master's Degree Program in Women and Gender Studies, Faculty of Social Science, Chiang Mai University, Chiang Mai 50200, Thailand
2 Institute of Sociology and Cultural Organization, Leuphana University, 21335 Lüneburg, Germany
3 Department of International Affairs, Payap University, Chiang Mai 50000, Thailand
* Correspondence: petra_l@cmu.ac.th (P.L.); anthony.waters@leuphana.de (T.W.)

**Abstract:** Promising research from Thailand already highlights women in the sexual entertainment industry as being active participants in both intimate relationships and commercial transactions simultaneously. Notably, they are neither victims nor alienated laborers, as some activist narratives assert. Women working in Thailand's sex entertainment industry consistently adapt working cultures to modernity's demand to reduce sex to a commercial transaction while often seeking emotional engagement. One result is that new forms of intimacy emerged, taking on new cultural meanings. The profoundly felt need to care for and take care of someone else [*dulae* (Thai: ดูแล)], seen as a form of "intimacy", is, in fact, deeply rooted in the Thai social context. We reframe the literature about sex work in Thailand by assuming that intimacy is key to understanding how "sex work" arose and is sustained there. Focusing on intimacy distances research about sex work away from western assumptions about the commodification and alienation of labor. This gives a more holistic understanding of the complexity of overlapping and intersecting dimensions of the work women perform in sex entertainment. "Intimacy" ties together the issues of money, labor, and a need to care for someone and be taken care of. This thread links women with their customers, families, and themselves.

**Keywords:** Thailand; sex entertainment; intimacy; emotional labor; care

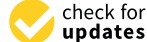



## 1. Introduction

Women working in the Thai sexual entertainment industry are typically stigmatized through ideologies of religion, cultural values, and economic interests. Such women in the sexual entertainment industry are often subjected to shunning and forced into marginal areas of society where they are more likely to be victimized by violence, addiction, and mental illness and labeled with dysphemistic terms like "prostitute". Because sexual entertainment is stigmatized and sometimes even illegal, women are less likely to seek assistance from police, social workers, and other professional services. In countries where prostitution is criminalized, they are subject to arrest, fines, and incarceration. Meanwhile, their customers, the consumers of sex products, businesses commodifying sex work, and the larger society served by prostitution and the sexual entertainment industry are often given a pass. In short, prostitution and sexual entertainment are embedded in patriarchal norms and laws. To a large extent, this results from the exclusion of voices of marginalized groups allowing for dominating powers to have "the privileged act of naming" and the power of framing that emerges from "interpretation, a definition, a description of their work, actions, etc., that may obscure what is really taking place," generating a false vision of reality (Hooks 1991, pp. 3–4).[1]

## 1.1. Sex Slavery and Liberation

In recent decades, feminist activists from the wealthy individualistic West have asked sharp questions about the consequences of shunning and have sought policies to mitigate the effects of the stigmatization in places like Thailand. General narratives developed to do this, often in opposition to each other, with others in-between. At extremes, these are the narratives of the "sex slave" and "sex worker". Both are used to favor preferred framing and public policies to address the "social problem" of prostitution.

On the one hand, advocates frame prostitution as coercive "slavery" in which the women are the victims of traffickers. The approach focuses on prostitutes as victims and advocates harsh penalties for traffickers, consumers, pimps, and others, while sending women in for counseling and treatment. In the case of Thailand, this perspective takes a radical view of sex work and points out that women are victims tricked into playing into their own repression (Farley 2004). Generally, such feminists see all commercial sex acts as patriarchal and oppressive. Such Feminists question whether sex work should be seen as work and use "paid rape" instead (see, e.g., Bindel 2017, 2020; Cawston 2019). Logically, they advocate banning sex work and the sex industry (Weitzer 2007) and assert that sex work causes harm and is always a form of violence against women (Gerassi 2015). The logical conclusion to such policies is that pimps and customers should be arrested, bordellos closed, and the prostitutes—who are victims—should be provided education, social workers, and training for "legitimate" employment. In Thailand, several programs undertake such an approach, starting with the Christian International Justice Mission (IJM), which defines most prostitution as slavery, and sends volunteer 'investigators' to rescue women. The investigators tip the police off to people they assume are trafficked. They also typically advocate for social work efforts and retraining programs that seek to have women complete vocational training.

## 1.2. When the Problem Is Identified as "Stigma"

In an attempt to eradicate human trafficking and the sexual exploitation of women and children, the Swedish Neo-Abolitionism model criminalizes all third parties involved in prostitution. It is based on the firm belief that all forms of prostitution are based on gender inequalities and are inherently violent acts against women. This approach aims to eliminate prostitution because it does not fit the moral standards of a social-democratic welfare state that emphasizes gender equality. Some women still working in Sweden are marginalized and stigmatized yet further as they are inherently seen as victims without a choice (Santana 2021, pp. 51–57).

Countries like Germany and the Netherlands legalized prostitution. This approach assumes away the very nature of stigma (e.g., Goffman 2009), assuming that since society, not the individual does labeling and framing, the approach asks, why is prostitution not just another job? It seeks to normalize sexual services as alienated "work" like any other profession and advocates for worker and consumer protection laws similar to those of other personal service workers such as barbers, beauticians, waiters, etc. The business and government then protect workers by providing the right to unemployment insurance, sick days, and paid holidays. The government also collects taxes while at the same time focusing on the safety of workers and their customers (Weitzer 2017). This reframes prostitution as sex "work" in a labor market, with the provision of sexual services as just another alienated product in the service economy, just like hairdressing, bartending, restaurant workers, flight attendants, and other occupations rooted in emotional labor (see Hochschild 2012b).

Although the legal framework provides sex workers with fundamental rights, it displays several drawbacks in terms of practicality. For example, many sex workers in Germany and the Netherlands are immigrants and unable to register with the authorities, leading to social exclusion. Additionally, registration requirements may not be effective because women still fear stigma and other repercussions, forcing them again into illegality (Santana 2021; Wagenaar et al. 2017). Furthermore, compulsory health checks to control

sexually transmitted diseases pose a considerable debate, especially amongst feminists, as it is seen as discriminatory and a violation of women's privacy.

New Zealand is the only country in the world where sex work is completely decriminalized in the "The Prostitution Reform Act of 2003". Although prostitution itself was not illegal, there are constraints hindering sex workers from legally performing their work in New Zealand (Santana 2021, p. 64). The emphasis on the decriminalization model lies in creating a safe working environment for otherwise marginalized sex workers. There is a high emphasis on self-determination and education rather than prescribed processes. This is especially apparent in health safety, where both customers and workers are encouraged to adopt safety measures, but this is not legally compulsory. Such decriminalization is widely seen as the most desirable standard for sex work amongst sex worker activists. In Thailand, for example, Empower supports this approach and sees it as less exploitative and prescriptive. They reason that power and decision-making are better with the worker instead with (patriarchal) legal frameworks. However, this approach also has its drawbacks, as again, migrant workers and illegal immigrants are still easily victimized and still face obstacles under this model (Santana 2021; Wagenaar et al. 2017).

Additionally, empowerment activists increasingly draw on the care-work framework, emphasizing the caring nature of sex work. This stands in contrast to opponents of prostitution, for whom the actual sex act is the main issue. Activists using the care-work frame highlight the variety of services women perform apart from sexual intercourse, using the concept of care work or emotional labor because care is widely understood as morally acceptable (Shih 2018). In Thailand, the NGO Empower,[2] founded by sex workers, is best known for this approach. There is, of course, much in between the extremes of IJM and Empower; however, these narratives frame discussions of Thai sex work, particularly with foreigners. What they share in common, though, is an emphasis on the commodification of sexual activity and the framing of sex work as a "social problem" demanding resolution through the application of social and political power.

*1.3. What about the Spaces "In-Between" That Offer Intimacy and Caring (Dulae)?*

The two contrasting views highlighted above reflect a paradox that sex work is simultaneously "empowering and exploitative, safe and unsafe; as well as a space that can involve free and restrained choice, while both resisting and reinforcing gender stereotypes" (E. M. Smith 2017, p. 346)". In contrast, the "social problem" approach using terms like "sex slavery" and "empowered women" are reductionist. The problem is that reducing the sex entertainment business to such binaries leaves little room for non-judgmental analysis. Most notably, individual women's location and cultural context are omitted when the argument is reduced by concepts like decriminalization, legalization, and criminalization of customers.

We believe that both approaches are overly "deterministic" because they reduce sex entertainment and prostitution to a simple sex act and typically draw on research rooted in western individualistic experiences. It leaves out the "in-between"; spaces in which resistance and negotiation of the self take place. Over the past two decades, research on sex work and feminist discussions have moved beyond the dualistic view. Emerging research already suggests that there is more to sex work than a simple black-and-white picture. Examples from Latin America and the Caribbean give a more nuanced description of the reality of women, often forced into prostitution through social and economic inequalities, growing poverty, and the hope for a better life with a foreign husband. (see, e.g., Brennan 2004; Carrier-Moisan 2020; Padilla 2007). Similar works exist for Southeast Asia, notably Lisa Law's (2000) study of bar workers in Cebu/Philippines, Heidi Hoefinger's (2014) analysis of Professional Girlfriends and transactional relationships in Cambodia, and Kimberly Hoang's (2015) study of Vietnamese hostesses. They also emphasize the dangers involved, the toll the work in sex entertainment takes on the body and mind, and the risk of contracting diseases, violence, and mistreatment. However, all of these anthropological studies also highlight women's subjectivity, their resistance to existing norms, and the

negotiation processes required to create authenticity, love, or intimacy within the encounters with their (Western) customers.

Nevertheless, particularly when looking into studies of sex work in Thailand, the 'in-between' spaces seem sparse. Most research emphasizes either the exploitative or liberating nature of sex work. However, such approaches miss the role intimacy plays between women and their "customers" in the go-go bars, karaoke clubs, massage parlors, etc., which market to foreign tourists. Such institutions appeal to the fantasies of foreign men for a compliant Asian woman while also providing a venue for the Thai woman to dream about an intimate relationship in which she can care for her husband, family, and children. We want to highlight that this is especially true when data and conclusions from Thai theses and books are included which are not accessible to a broader audience. Thai authors are already covering "in-between" spaces, moving aways from the prevailing narratives and victim/liberated women dichotomy.

This analytical essay reframes the literature about sex work in Thailand by assuming that intimacy is key to understanding how "sex work" emerged and is sustained there. As discussed, in Thailand, such intimacy extends beyond the sex act. It may include not only the customer but also non-sexual relationships with the woman's children, parents, extended family, and friends, but also themselves. The Thai word *dulae* (Thai: ดูแล) summarizes how this intimacy is bounded in the Thai social context. *Dulae* describes the need to care and take care of someone else with mutual, often life-long obligations and is at the heart of the need for a human connection, summarized most precisely by the English word "intimacy". This applies not just to romantic partners but also mothers and their children, parents, and others.

Intimacy does not fit into the implied "either-or" approach usually applied in research in the Thai context. It leaves out aspects necessary to understand prostitution in a country like Thailand, where sexuality and intimacy express themselves in ways different from that typically seen in activist narratives. Or, as Udayagiri (1995, p. 163) states: "[w]ithin such a binary analytic, then, women can only be emancipated through Western economic rationality," which in the case of Thailand seemingly means choosing between policies that criminalize "sex slavery," police crackdowns on women and brothels, or emancipatory policies which emphasize legalization and destigmatization. Therefore, paying a closer look into original Thai research and literature will help to understand the nature of sex work in Thailand, with intimacy in its heart.

## 2. Statement of the Problem

Many of the studies of sex workers in Thailand start with an assumption that there are discrete entities and definitions of who people are and what they do. Much of this happens because sex work is often defined in terms of legalistic formula, using pre-existing categories like prostitute, sex slave, customer, pimp, and a range of definitions borrowed from the western literature. In court, individuals are forced into the language of guilt or innocence. Moreover, the language of the marketplace places sex "work" into the marketplace, in which "sex work" is service labor in exchange for money, or in other words, alienated labor is assumed to be separable from the worker, just as it is on a factory line. This, indeed, is the language of western derived social sciences. In such thinking, there is little room for an "in-between" rooted in emotions.

But what about the case in Thailand, where the norms are different? Emotions are at the heart of the Thai philosophy that emerges from Buddhism, where the point is to suppress such emotions on the path to seeking Nirvana. The result is that when western NGOs frame sex as "work" and especially as a social problem needing to be solved by suppressing slave traders and pimps like IJM does, or campaigning against the stigmatization of sex work in the fashion that Empower does, the question of how the women themselves see their world is missed.

The problem with the Social Problem approach to sex work in Thailand is that it leaves out the intentionality of the sex worker herself. This is seen especially in the Thai literature,

where scholars describe women who have a life outside their occupation and have dreams and goals that involve "caring" *dulae* and intimacy, even with "customers". The women do not simply frame their work in entertainment venues in terms of alienated labor but talk about their own emotions, including the type of men they prefer (and reject), friendships, their own children, and larger families. They are searching for an intimacy that will serve their goals of taking care of both themselves and their family, financially and emotionally.

Intimacy can also be perceived as part of the general shift from traditional to modern modes of life. In Pre-modern times daily tasks and life, in general, followed moral guidelines based on local traditions. Responsibility to "care" *dulae*, which were taken for granted and communicated informally from the past. New values and forms of living were introduced with the arrival of modernity. The new values stood in stark contrast to the pre-modern or traditional societies like Thailand's rural Northeast (see Lapanun 2018), where poor, isolated farming communities persisted until recently outside the global order, with its challenges to traditional morality.

Notoriously, the move from primarily poor rural areas of Thailand to cities meant a step towards more modern market-based habits, irrespective of whether there was coercion, free will, or some combination of the two (Khamphouvong 2019). Modernity also opened new forms of living for the traditional family and the intimate non-sexual relationships that persisted there. This development deeply affected women as traditional norms for motherhood and childrearing changed. Kinship and the rights and obligations that traditional sexual relationships encompassed changed too. Patriarchal and heterosexual norms shifted, offering women a new, more liberated outlook on sexuality

The heightened interrelationship between the influences of globalization, capitalism, and "individualism" markers of modernity (see Hofstede et al. 2010, pp. 18–20, 450–52). While the country and its people internalize traditional values, increasing globalization catalyzes new forms, mixing external norms brought from the global marketplace with older norms for intimacy in family relationships, gender, and hierarchy found in Thai society. Intimate relationships, sexual and otherwise, became a continuous negotiation and reflection process based on the current context, including values of the marketplace, and were more than norms inherited from ancestral traditions.

## 3. Research Aim and Methodological Approach

This analytical essay is based on an extensive literature review about sex work, focusing on the "in-between" spaces, often neglected in the context of Thailand but which are more apparent in original Thai sources. We aim to highlight the essential role of intimacy within sexual work encounters by looking into the existing literature on sex work in Southeast Asia, particularly Thailand. We translated selected passages from original Thai research to underline our argument. Additionally, Thailand's historical development, cultural norms, values, and rituals shed light on a less-regarded aspect of work in the sexual entertainment "industry".

## 4. Theoretical Framework

### 4.1. Intimacy

To understand how intimacy is created, we first need to establish a working definition to further develop its meaning in Thailand. Simply put, intimacy refers to the physical and/or emotional closeness or familiarity within an interpersonal relationship. It is often associated with closeness in sexual relationships between partners, and the highest level of intimacy is reflexively assumed to be sexual intercourse. However, intimacy occurs in other relationships that are not generally sexual. Examples include an attachment to family members such as children and parents or even relationships with good friends.

The most commonly known concept of intimacy is the "Interpersonal Process Model of Intimacy" (IPM; Reis and Shaver 1988). "According to this model, intimacy increases when an individual discloses their personal thoughts or emotions (i.e., self-disclosure) to

their partner, the partner responds supportively, and the partner's response is perceived by the discloser as validating and caring" (Khalifian and Barry 2020, p. 59).

Jamieson (1998, 2011) questions this rather simple model and the significance of mutual self-disclosure in modern, intimate relationships. She highlights that "love and care expressed through actions is a very different dimension of intimacy . . . ". The daily practice of complementary gifts, e.g., the man's wage and the women's housework, symbolize mutual care and love (Jamieson 1999, p. 485). She highlights that there are culturally different but transferable concepts which she coins "practices of intimacy" These "'practices of intimacy' refer to practices which cumulatively and in combination enable, create, and sustain a sense of a close and special quality of a relationship between people" (Jamieson 2011, p. 3). Practices of intimacy do not necessarily involve intimate sexual acts, which accounts for the concept's ambiguity and its possible intersection with related patterns of feelings and inclinations "such as trust, empathy and respect" (p. 3).

Moreover, Jamieson (1999) challenges the underlying notion of modernity's shift to "pure relationships" marked by equality among partners (Giddens 1992). She gives a more nuanced picture of the reality of intimacy, questioning increasing equality, and highlighting power and gender inequalities inherent to the concept. She moves away from the too optimistic and simplistic outlook, stating that "[e]mpirically, intimacy and inequality continue to coexist in many personal lives" (Jamieson 1999, p. 491).

We acknowledge that gender inequalities are apparent in the context of Thailand. Furthermore, there are power imbalances based on education, ethnicity, gender, and economic status. This is especially apparent in the often inequal (economic) power relations between women working in sex entertainment and their customers. However, we want to use the intimacy frame and "practices of intimacy" as a basis to help understand how the Thai concept of care *dulae* is key to the work in sex entertainment and how it works on different, often overlapping dimensions, such as society, family and friends, customers and also the women themselves.

### 4.2. Emotional, Affective, and Alienated Labor

Modernity and the change in work structures also highlighted the importance of new forms of labor. Feminist theorists, in particular, have long been concerned with immaterial and affective labor. Gender inequality, reflected in the unequal distribution of unpaid care and domestic work between men and women, is a significant feminist issue. Arlie Hochschild (2012b) described how, in the 1970s and 1980s, "emotional labor," whether that of flight attendants, bill collectors or parents, as "work" not completely commodifiable in the labor market. She pointed out that alienated emotional labor took a toll on the flight attendants and bill collectors she observed, including changes in personality, anxiety, drinking, and other undesirable behavior. In her later work, *The Outsourced Self*, Hochschild (2012a) concludes that capitalism has commodified everything, including intimate aspects that were previously taken care of by family members or a close community. She highlights that people in modernity constantly need care from an expert team of skilled workers they hire and pay to subsidize their own lack of skills.

Critiques of Hochschild question her deterministic stance on emotional labor and its inherent alienation (see, e.g., Bolton 2005; Hardt 1999; Lazzarato 2006). Sharon C. Bolton (2005) questions that capitalism took over and "appropriated all of our feelings so that there is no longer any room for sentiments, moods or reactions that have not been shaped and commodified via the 'commercialization of intimate life'" (p. 2). She further criticizes Hochschild's view on employees as she "creates an illustration of emotionally crippled actors" (p. 48) who presumably are no longer able to act. Lazzarato (2006) uses the term "active subjects" to emphasize the dynamic nature of its [affective labor] production" (p. 134).

Furthermore, modern working environments make it "increasingly difficult to distinguish leisure time from work time. In a sense, life becomes inseparable from work" (Lazzarato 2006, p. 137). This is where former theories on emotional labor are inadequate

as guides for future application. These concepts base their analysis on the assumption that there are separate spheres and a high possibility of the self becoming alienated labor. Seemingly, in modernity, differentiating between one's self at work and the self in other aspects of life is no longer possible. "The subjectivities shaped at work do not remain at work but inhabit all the spaces and times of nonwork and vice-versa" (p. 246).

Since work in sex entertainment is placed in the service sector, many scholars utilize Hochschild's concept to examine its effect on women, acknowledging the factor 'emotional labor' as key (see, e.g., Brents and Jackson 2013; Chapkis 1996; Sanders et al. 2018). Some of them emphasize sex work's alienating nature and sex workers' strategies to create emotional or physical boundaries. They further highlight the negative consequences of extensive emotional labor, such as drug or alcohol abuse and other mental health issues (see, e.g., Abel 2011; Sanders 2002, 2004; Senawong 2019). Then again, other scholars move beyond dualisms of the self at work and self in life. They acknowledge that women working in sex entertainment have fluidity and a hybrid self that moves between different spheres. These women utilize emotional labor as a strategy or tool to create love, intimacy, and desire, without clear boundaries (see, e.g., Brennan 2004; Carrier-Moisan 2020; Hoang 2015; Hoefinger 2014; Muangjan 2018; Weerakulthewan 2000).

## 5. The Sexual Entertainment Industry in Thailand—A Review

### 5.1. Traditional Role of Thai Women in the Household

Women in modern Thailand increasingly enter the public sphere in business or politics,[3] yet "women in Thai culture also still hold the central position within the household structure, which requires them to be involved directly with matters of the home by being both a dutiful wife and a mother, along with holding down a job" (Avila 2008, p. 12). Such burdens are rooted in moral and ethical traditions, felt by women in high-status positions like doctors and stigmatized professions like sex workers. Females in traditional Thailand are seen as the nurturers of the family, with caring attributes essential for a functional family (Wongboonsin and Tan 2018). This includes norms for the appearance of sexual purity. Not maintaining such norms meant "one may risk being labeled as bad women or similar to prostitutes," a stigma that could affect the family as a whole (Skulsuthavong 2016, p. 28). Avila (2008) highlights that women "are expected to be reserved and demure, refrain from showing interest in men, and uninterested in sex since prostitutes are exactly the opposite" (p. 28). "Gender double standards" remain strong in modern Thai families. "Sexual expression and experience were acceptable for young unmarried men, but not for young unmarried women" (Tangmunkongvorakul et al. 2011, p. 332). Even today, young women have an ambivalent attitude towards sexuality outside marriage due to the persistent gendered double standards. Although in modern Thailand, women have more liberty and a higher partner turnover than before, "they experience cultural [traditional] values that inhibit their sexuality outside marriage and open discussion with adults about sexual matters" (Tangmunkongvorakul et al. 2011, p. 335).

In the traditional Buddhist Thai family, men conventionally are the symbolic head of the family and occupy the most important religious posts as monks. Women still play an active and vital role in rites (Watson Andaya 2006, p. 328) and are even dominant in many household matters. A similar gendered division of labor meant that while men were symbolically dominant, women made many day-to-day decisions, and "Female prominence in economic life, [is] still a feature of Southeast Asian societies in contemporary times" (p. 329). To some extent, this reflects traditional norms that put the mother at the center of family life, including controlling household finance. In societies such as traditional Thailand, the occupational role of a woman is intricately connected to her overall status within the community: the higher her economic conditions, the higher her status (Yoddumern-Attig et al. 1992, p. 83).

Ideal women were both dutiful wives but, before that, dutiful daughters who *dulae*. "Care and support of elderly traditionally rested with the family through a system defined mainly in terms of filial obligations of adult children" (Knodel et al. 2013, p. 129). Parents

and their children are bound by social and cultural principles rooted in Buddhist duties toward the parents. The ethical reciprocity of the "debt of gratitude" *bun khun* (Thai: บุญคุณ) "place[s] all children in a position of life-long obligation to their parents" ([Mills 1999](#), p. 76). Moreover, although both son and daughter hold the same commitments, a boy can generate "a store of merit for his parents" by entering the monastic order, even for a short while (p. 78).

Women are barred from any form of monastic life that would count against their filial piety, so they have to fulfill their duty differently. "[A] daughter can uphold her *bun khun* obligations to parents only through her respectful obedience to their authority and by her contributions toward the physical and material well-being of the [parental] household" (p. 78). This obligation continues after marriage, and the mother's central role in Thai cosmology is recalled each Mother's Day when children prostrate before their mother and apologize to her for the pain they caused during childbirth and after.

Such deeply felt obligation continues after marriage and departure from the natal home. Indeed, the marriage of a daughter to a high-status family is important for her parents' status; it gives her the capacity to continue assisting with their physical, material, and emotional needs. This was particularly the case when a girl "married up" to a high-status man, whether out of duty to the parents, love for the man, or a combination of both. Relationships with both parents, and a man assume intimacy. The first is rooted in the context of filial piety, and the latter is sexual. However, both are intimate.

### 5.2. Marriage and Sexual Infidelity

The woman's role as a mother is central to Thai culture and, in some respects, trumps that of a wife. The term "mother" *mae* (Thai: แม่) marks respect, appreciation, and also esteem and can be found in many Thai expressions such as "river" *mae nam* (Thai: แม่น้ำ), highlighting the importance of the source of life. Women who give birth and raise their children are highly respected and are "considered natural community leaders" ([Yoddumern-Attig et al. 1992](#), p. 25). Women ideally seek a stable relationship with a faithful man who provides for their economic and emotional needs; men want their partners to "be good housewives and good mothers" ([Knodel et al. 1997](#), p. 297). This form of intimacy between marital partners is deeply reciprocal, and taking care *dulae* of one another is a crucial aspect of interpersonal relationships in Thai culture. But marriage also is traditionally perceived as a bond between two families rather than just between two individuals, as highlighted by the phrase *taengngan kap khropkhrua* (Thai: แต่งงานกับครอบครัว), "which literally translates to 'marrying the family'". Romantic love, generally regarded as a by-product of capitalism and individualism with 17th and 18th-century European roots, "has been heavily popularized and spread via Western-dominated films and media," reaching Thai adolescence in urban but also rural settings. There is an increased understanding that "romantic love and partnership is predicated on emotional connection between two people; indeed, attaching the local value of sexual restraint to the more global value of romantic love enables the maintenance of this cultural value even in a more globalized setting" ([McKenzie et al. 2021](#), pp. 14–15).

Opinions differ significantly regarding sexual fidelity within a marriage. Studies conducted in the 1970s showed that the infidelity of married men, "limited largely to [nobility], wealthier farmers and upper-class urbanites," took the form of concubinage with men having several "minor wives" *mia noi* (Thai: เมียน้อย) living separately from their "first wife" *mia luang* (Thai: เมียหลวง) ([Knodel and Prachuabmoh 1974](#), p. 440). Such polygamy was legal until the 1930s.

But relationships for lower classes that involved acknowledgment of second families were traditionally abhorred. It was in this context that commercial sexual encounters between married men and prostitutes were tolerated by the wives as opposed to "an affair with an 'ordinary' girl or, even worse, taking a minor wife" ([Knodel et al. 1997](#), p. 300). The occasional visit to a brothel did not threaten the family's core as it was believed to involve neither financial nor intimate emotional commitment, at least in the view of Thai



wives. Thai generally practice marital monogamy today, but multiple relationships are common even today; there is a modern word *kik*[4] (Thai: กิ๊ก) for such outside relationships, which can be applied to both men and women. Male sexual infidelity is a widely acknowledged fear, though not necessarily stigmatized. Marital infidelity is stigmatized for wives (Knodel et al. 1997, pp. 298–99).

### 5.3. Thai Women, Foreigners, and the Sexual Entertainment Industry

In this traditional context, lonely American soldiers assigned to air and naval bases in Thailand during the American War in Vietnam sought intimacy and companionship modeled on American dating culture. The concept of the "rented wife" *mia chao* (Thai: เมียเช่า), became common around military bases. This typically meant a temporary agreement between the Thai woman and her military "customer/husband" to spend time semi-exclusively with each other. Such relationships were often marked by deeper emotional attachment and not always a pure money-for-sex transaction (see, e.g., Jaisuekun and Sunanta 2021; Lapanun 2018; Wongsavun 1973, 2022, pp. 50–55). When the American soldier inevitably finished their deployment in Thailand, the breakups were often painful, particularly when children were involved.

Although international and transnational relationships in the context of Thailand existed way before the American GIs flocked to the entertainment areas, these new-style intimate relationships between Thai women seeking to honor parents, and wealthy American men seeking a girlfriend-type intimate companion, provide an important template for the cultural tensions that emerged and continue to play out in the sexual entertainment industry of Thai tourist areas like Pattaya, Patpong in Bangkok, and Phuket. Thai women sought to marry "Western foreigners" *farang* (Thai: ฝรั่ง) to meet the expectations of the family, and in this context, the term "foreigner's wife" *mia farang* (Thai: เมียฝรั่ง) became a synonym for both shame and pride. Women from Thailand's Northeastern *phak-isan* region (Thai: ภาคอีสาน) moved to the areas around the military basis to work in the sex entertainment businesses and perhaps find a Western husband. These relationships were seemingly based on a mutual desire for the "foreign" or "exotic". "While men seemed drawn to Asian women by the promise of [what farang considered] 'traditional values,' Asian women were often attracted to Western *farang* men and societies because of their assumptions about 'modern' ways of life and more flexible gender relations in Western countries . . . " (Lapanun 2012, p. 7). The motivation behind these relationships is a desire to "break away from traditional gender roles and expectations" for Thai women, while *Farang* men can "reaffirm the sense of traditional masculinity" (Jaisuekun and Sunanta 2021, pp. 145–46).

This is why, as Thompson et al. (2018) note, reducing these relationships to "mere financial and material exchange" is a mistake (p. 109). *Farang* men contribute in other forms to the Thai wife's family. The "caring" *dulae* for the (existing) Thai children of the wife, and supporting parents and family members, are driven by the understanding of traditional gender roles—e.g., being a "real man" (Jaisuekun and Sunanta 2021, p. 146)—which is not often available in their home countries. Successful men will extend this role by being active members of the local community, and involvement with Buddhist traditions and rituals. The couple often open restaurants or bars focusing on foreign customers who play "an important role in connecting and localizing *farang* and their transnational married life with Thai village women to the rural Isan society" (Thompson et al. 2018, p. 114). By marrying a foreign man, some women, particularly in the Isan Region, strengthened matrilineality, bringing honor to the family (p. 105).

### 5.4. The Emergence of the Thai Sex Industry

Until 1960, prostitution in Thailand was legal under the Contagious Disease Act of 1908. Brothels and prostitutes registered with the government and paid a fee to operate or work. In 1960, the Prostitution Suppression Act was introduced, which made prostitution illegal in Thailand, and women were arrested for offering sexual services. This was before

significant international tourist demand emerged; previously, Thai men were the main clientele, often in the context of concubinage traditions in which high-status men legally and extra-legally took multiple wives. The emergence of bordello-based prostitution allowed middle-class men to imitate the habits of their social superiors for at least a short time.

In 1966, the Thai government introduced the Entertainment Places Act to regulate forms of entertainment places like nightclubs, bars, and massage parlors designed to serve American soldier's cultural preferences for sexuality expressed in go-go bars, night clubs, where there was an allure of a "date" like relationship. "This 1966 Act set the stage for an agreement with the U.S. military to allow American soldiers stationed in Vietnam to come to Thailand for rest and recreation" (Singh and Hart 2007, p. 162). These laid the groundwork for the modern Thai commercialized sex entertainment businesses.

"In this context, Thailand's sex entertainment businesses took on new forms. Today, the tradition means "[s]ex workers are often legally employed as entertainers in venues, such as karaoke bars, beer bars, a-go-go bars, and "massage parlors" *ran nuat* (Thai: ร้านนวด)[5] where providing sexual services occurs supplementary to entertainment work".

The most common concept is go-go bars, where women animate customers to drink and buy them drinks.[6] This model drew on western norms regarding the roles of socializing, flirtation, and "one-night stands". So-called "leisurescapes" emerged, or as Gkoumas (2022) calls them, "sexscapes as leisure-oriented spaces of heterosexual sexual conduct in non-tourist areas in Thailand". That goes in line with Elizabeth Bernstein (2007), who sees a "profound transformation in the erotic sphere" in the postindustrial era (p. 6). Sexual activities became increasingly mixed with recreational ones, creating a new form of sexual ethics. "Instead of being premised on marital or even durable relationships, the recreational sexual ethic derives its primary meaning from the depth of physical sensation and from emotionally bounded erotic exchange," which Bernstein coins "bounded authenticity" (p. 6).

In this context, sexual services are supplementary to women's entertainment work and occur only in private places or hotel rooms.[7] Brothels were transformed into bars catering to local entertainment demands as well. "A larger segment of the entertainment sector [also] serves local clients" (Villar 2019, pp. 111–12). Indeed, the Thai government continuously promotes "intra-Thai tourism, which arguably tends to promote the sex sector" (p. 112).[8]

It is hard to find reliable numbers of sex workers in Thailand due to the trade's illegality; after all, there is no discrete "sex worker" category in the Thai labor reports, which is not surprising given the entanglement with non-sexual entertainment activities. Neither is "search for intimacy" readily classifiable as "work" or "relationship" even by the most creative survey taker. Nevertheless, in 2016, the Thai government estimated that around 300,000 people worked in the sex entertainment business, a figure spread by the Empower Foundation (2016), the advocacy organization supporting sex workers' rights in Thailand.

Who are the Thai women working in Thailand's sex entertainment? Although there are no precise demographics and the reasons to enter this industry are manifold, there is a collective understanding that women, mostly single mothers[9] from more rural areas of Thailand and neighboring countries, move to bigger cities to serve tourist demand for sex entertainment to help their families (Chia 2016; Ouyyanont 2001).[10,11]

*5.5. The Problem of the Victim/Liberated Woman Dichotomy*

The discourse about work in Thailand's sex entertainment is rooted in both ideological and practical voices. Even in Thai literature, the "Madonna/whore dichotomy" finds its way to the public, defining what a "good" and "bad" woman is: sexual restraint and motherhood as opposed to "sexually adventurous, promiscuous, and unrestrained" (Feangfu 2011; Harrison 2000). In the context of prostitution, women in Thai literature who are driven by circumstances only gain a form of pity and an acknowledgment that they might be "victims of a wider social evil," but not more.[12] Much of this assumes the primacy of the individual. There is no critical analysis of the role of Thai females in society in general,

"and female prostitution in particular," let alone from a feminist perspective (Harrison 2000, pp. 184–85). It is not surprising that the current English discourse on sex work in Thailand, much of it from activist NGOs seeking contracts and donations, balances uneasily between an assumption that sexual entertainment is either oppressive or liberating.

### 5.5.1. Freeing the Slaves—Women as Victims

Particularly in Thailand, advocates emphasize human trafficking and forced labor by women and children as victims who enter involuntarily work in sex entertainment through dubious job opportunities and parents unknowingly selling their daughters into the sex trade (T. A. Smith 2019, pp. 18–19). The victims are usually women "who look for better employment, in need to support families . . . " and are therefore an easy target for recruiters who search for victims nationally in the surrounding countries, and Thailand's impoverished "hill-tribe areas" (Opanovych 2016, p. 106). Others from Myanmar and other nearby countries enter Thailand illegally. They often do not speak Thai well and are easily exploited by traffickers.

The United States has sex trafficking at the center of its foreign policy and funds NGOs,[13] such as The Asia Foundation (TAF)[14] and other entities, to combat human trafficking and sex slavery. NGOs[15] operating in Thailand aim to free women from inhuman and oppressive sex work, as there is the general assumption that they are victims with no choice. They are then encouraged to choose "jobs considered appropriate" and "sew or bake cookies instead" of sex work (Chandran 2019). Typically, in the activists' minds, this means factory work in the vast network of Thai garment factories and small retail street businesses, entailing long hours, low wages, and elevated risk.

The sex slave frame reflects American concepts of individualism, agency, and labor. The notion behind this external support is that Thailand cannot solve this issue independently without (Western) support. One of the main objectives to stop women from falling victim to "sex slavery" is, they reason, to stop sex tourism completely as "it appears to be the core of the problem" (Opanovych 2016, pp. 108–9). In response, many nations, especially the United States, prosecute their citizens who engage with child prostitutes, particularly in Cambodia and Thailand. The general idea is to deter American men from such activities and decrease demand.

But the human-trafficking narrative gets even stranger when some NGOs offer so-called "reality tours" to sex entertainment areas to provide voyeuristic customers a real-time experience. These tours aim to highlight the importance of supporting NGOs, paired with the willingness of tourists to gaze at people offering sexual entertainment services, presumably in a detached, amoral fashion. Many "reality tours" organizers are American Christian organizations that otherwise discourage young men from visiting sex entertainment venues (Bernstein and Shih 2014).

We do not underestimate the role of human trafficking in sex entertainment, as it happens to many women, although not all. We acknowledge that forced labor in any sector, not only sex entertainment, is a human rights violation that should be prosecuted and punished. The problem is that when prostitution gets conflated with human trafficking, it makes a more analytical view difficult and ignores the many women that voluntarily and consciously enter sex entertainment in Thailand.

### 5.5.2. The Empower Approach—Sex Work as Labor

Acknowledging that sex work has a liberating and empowering aspect is also part of the "sex work is work" movement. In Thailand, this is spearheaded by Empower Foundation. The "work" model they advocate, assumes that sex work is alienated labor devoid of intimacy, like any other kind of service work. Testimonials assert: "This is what we choose to do, and we feel a sense of pride and satisfaction that we are just like other workers" (Chandran 2019). Women employed or associated with Empower are mainly concerned with insisting that their work be recognized and decriminalized. They seek to ensure a safe working environment with less stigma and pressure.

Yu Ding (2020) recognizes scholars' efforts in promoting "women's free will, right to labour and agency" through the concept of "sex worker". However, she highlights that adopting this Western concept uncritically is problematic. "Without looking into the women's own perceptions of 'sex work,' what they do and aspire to do in a specific context that is socially, economically, culturally, and historically different from the West, our understanding of prostitution is incomplete (Ding 2020, p. 95).

This goes in line with Lapanun (2013), who agrees with the "women's agency approach" but also highlights its limitations because it solely concentrates on the money-for-sex exchange without taking into account "the possibilities of intermingling between sexuality, money, and intimate relations that might lead to a long-term commitment between women in the sex industry and their clients (Lapanun 2013, p. 127).

### 5.5.3. The Thai Feminist Approach

Emerging Thai research contradicts much of the victim narrative and highlights Thai women's decisions to enter sexual entertainment. Instead of choosing a menial job in a factory or staying on the farm with their families, they try their luck in bigger cities. These women then need to negotiate with their "employers" and their (traditional) families, but also with themselves, finding ways to balance their lives in the city and their former lives back home. Nicolas Lainez (2020) called such negotiation processes "relational work". He emphasizes that people do not blindly follow ancient cultural scripts. Instead, families draw on their understanding of "filial responsibility, duty and sacrifice" to create "relational packages" which consist of distinct "combinations of relations, transactions, media and negotiated meanings". These packages "allow families to negotiate controversial practices," including accepting money to pay debts through their daughters' sex work (p. 7). McKenzie et al. (2021, 2022) use a "relational choice" to refer to family-based decision-making processes that have the greater good, e.g., the family's well-being, in mind.

Persistent social structures and norms require this continuous negotiation, even as working in sex entertainment remains stigmatized in Thai society (see, e.g., Chandran 2019). This influences how women perceive themselves and their work and manage their identities. This reflects Songsamphan's (2004) view, who states that "[f]or many women, work in the sex industry is extremely empowering . . . " (p. 87). Still, such language is that of the alienated laborer; it does not reflect the broader language of intimacy.

However, it is not only economic factors that make this job for women attractive. For example, Weerakulthewan (2000), in her study about "women working in beer bars" *sao ba bia* (Thai: สาวบาร์เบียร์), highlights that these women often see their work as liberation from traditional Thai norms shaped by the societal stigma associated with the sexual activity of women. It further allows for liberation from traditional expectations of the role of women in the family and globalized society.[16]

The women she interviewed differentiate their work from simple "money-for-sex" transactions and prostitution as they can negotiate power relations and seek a level of intimacy, which they believe is seldom possible in conventional relationships. With time, they gain experience, skills, and confidence to deal with potential "customers" and bar-owners. Unlike traditional Thai women, these women approach men and negotiate a close or distant relationship on their own terms, which may or may not include sexual services. This reflects what Spitzer (2021) writes about the role of intimate labor in Thai beer bars. "When we come to work, we have to make ourselves beautiful and talk sweetly to the customers by asking them to purchase our beer. Also, we have to make jokes with them to be able to sell a lot of beer" (p. 914).

Muangjan's (2018)[17] research highlights that working in sex entertainment has value for these women and allows for freedom that can lead to contentment. This freedom includes choosing suitable customers with whom they feel attractive. But it also gives them a sense of freedom from Thai social norms while also permitting the women to return *bun khun* via sentiment and monetary gifts. Finally, it gives a new meaning to sexual relationships and an understanding that a woman's body does not belong to one man

only, giving them subjectivity rather than being a mere sex object. Through their work, the women Muangjan describes gain a form of agency that they would not be able to experience if they had stayed in conventional jobs, even as they remain part of the larger family.

Muangjan (p. 121) describes the views of a woman who worked at a snooker club.

> . . . there is a new value and meaning to the term "Service Worker". Looking at prostitutes as simply under-educated victims is not enough. It leads us to overlook the value of such work in the context of the very new meaning of the term "Service "Worker". Prostitution is more than about staring at a body; there is a new meaning, an almost meaning as with an "Illusion" that is indeed the new Service Worker.

> The identity created is a new thing, not that of simply "sex worker". For example, in the case of Wunsen[18], she meets her clientele at the snooker table. She does not seek a clientele for sex, she is not building "customers". It is different because they are seeking a service. Wunsen says, "The customers at the snooker shop are not buying sex". For the women, it is first a story of "communication".

> The women explain that you are not picking up customers to simply provide sexual services because we know each other. And even if we don't know each other "we become close in a particular way, by hanging out together, eating together, watching movies. We get money from the men to pay for our rented accommodation and to use here and there".

> [One woman], Wunsen sees it this way, "We do not sell sexual services because the people we go with are our friends, in an uncommitted relationship (*kik*[19]), not customers". And even after work, there is reason that "we go with them because we like them, and the man gives us money to use". Wunsen explains that she goes off with her beloved, but "every time we go off, it is not only about sex. It might be going to karaoke, eating, and watching movies without 'sex' as the main thing".

The relationships described by Muangjan between the women and their customers show "practices of intimacy" (Jamieson 2011) in which the customer provides monetary support while the woman extends her care and emotional labor in the form of entertainment and companionship. In the context of Thailand, this "practice of intimacy" translates to *dulae* (marital) relationships of mutual care that may include a legal or common-law marriage.

*5.6. Sex Entertainment—A Cross-Cultural Experience and Othering*

Much of the sexual entertainment industry in Thailand, with its roots in tourism, is rooted in an implicitly cross-cultural model. Most obviously, the women fulfill the image of exotic "Thainess" their foreign customers expect. Statements like these reflect the women's awareness of the distinction between the Thai and foreign *Farang:* "I know what Thai men like, and I can tell what they think, so I can satisfy their needs and desires. And most importantly, I understand what they say because they are Thais like me. Working for three years in Bangkok with many foreigners and tourists, it was very difficult because my English is not good" (Gkoumas 2022, p. 172). Still, the sex entertainment businesses in Thailand emerged from the Vietnam era, using a business model framed by the Western male imagination of Thai femininity. Women developed hybrid identities to navigate through intercultural differences and expectations. The hybrid identity emerges as a cross between idealized aristocratic Thai privilege and equally idealized traditions of American girlfriend-boyfriend relationships from the 1960s.

Relevant, perhaps, are tourism studies discussing how tourists create an idea of the place they will visit and tour. In doing this, they are "customarily provided a representation of 'the Other' that they encounter" (Bishop and Robinson 2000, p. 191). In places like Thailand, which market low prices, there are also differences in income embedded in what John Urry (1990) and (Urry and Larsen 2011) called "the tourist gaze". This is, of course,

part of the tourism experience in Thailand, where wealthy Westerners, Japanese, Arabs, etc., arrive seeking sexual encounters. The commodification of sexual experiences and the marketing of "exotic," understood as "non-Western," created a form of "entitlement" in foreign men visiting Thailand's sex tourism places (Bishop and Robinson 2000, pp. 191–92). Existing (internet) guides to sex tourism in Thailand created by travel agencies, business journals, and personal blogs describe sexual encounters. Even airlines like Air Asia sexualize Thai women, depicting them as readily available, beautiful, "exotic," and willing sex partners eagerly awaiting sexual encounters with wealthy Western males (Truong 1990).

The question is, how do women positively negotiate their sexuality, identity, and work in encounters with Western tourists and expatriates and still create forms of intimacy (Hoang 2015; Hoefinger 2014; Law 2000; Muangjan 2018; Senawong 2019)?

A good description of how a Thai woman might negotiate the intimate relations between not just customers but also her family is described in a narrative from 1973 in the book *Sattahip* by Rong Wongsavun. In the book, one of his informants, a "rental wife," is interviewed about the emotional context of her work and life next to the American naval base at Sattahip:

Rong: "I heard that some Americans . . . they get sudden assignments and have to leave—and if they already paid for the month as agreed, then they would transfer [the rental wife] to his friend".

Rental Wife: "I heard the same thing, too, *Pee* [older brother]. He probably negotiated —to get something in return. You know that Americans will not give something away that easily".

Rong: "Have you ever been involved with that?"

Rental Wife: "I've never been...whatever their story...if you're gonna go to hell or wherever...and wants to give me to this guy or that guy —I will never go! Even if I'm bought like an object —like they said. I have honor, too, *Pee*—crazy! I'm not a piece of soap—use half of it and throw it away by giving it to someone else!"

Rong: "I'm sorry . . . "

Rental Wife: "You don't have to be too polite with me, *Pee*. I'm not a high madam from somewhere —only a madam from a hotel".

. . .

Rong: "You and your friends ever fight over husbands?"

Rental Wife: "We're women, *Pee*, and being the kind women we are —there's gonna be some fights—it's normal. . . . Believe me, *Pee*...if my husband gets bored of me, and gets another girl to be his wife—I lose my honor—I cannot explain it, but I will be so ashamed—I cannot look anybody in the eyes. . . .

. . . . Like I told you, "I have 2 small red-headed boys that I left with my mother to take care of. If I think about the future—it is about them. I try to make a monthly savings for them. Saving for them just in case something happens to me. When I came back this time, I went to the temple and promised to Buddha that I will not play [the gambling game] black frog/red frog anymore. I will do everything for the sake of money and for the sake of my sons." (Wongsavun 1973, p. 111. Translated by Siamrad Maher and Tony Waters)

Here, the "Rental Wife" highlights her negotiation power to balance the caring *dulae* demands of her customers and her family back home. Furthermore, she emphasizes her subjectivity as a woman and mother.

Fifty years later, women still show their subjectivity in the context of sex entertainment. Although the rented wife concept from the US naval bases moved to more contemporary terms like "entertainer," "offering company," or providing the "illusion fantasy of a girl not being a professional" in bar settings (Weitzer 2021, p. 644). However, the core of her life

remains the same: balancing the *dulae* responsibilities at the workplace with patrons, and caring for families back home (Gkoumas 2022; Spitzer 2021; Weitzer 2021).

Shih (2018) suggests that in Thailand, sex work is a form of care work, in the sense that through their work, women support and take care of their family and others around them. She quotes activist Malee: "If you look at sex work within an eight-hour day, only about five minutes is actually spent on sexual intercourse, the rest of the seven hours and 55 minutes, is about all the different kinds of work that we do—caring for everyone from our clients to our families" (p. 1084).

### 5.7. Affective Emotional Labor in the Capitalist Marketplace and Feeling Rules?

The traditional theoretical assumption is that women in Thailand's sexual entertainment business respond to consumer demand by constructing their bodies and themselves to appeal to their potential clients. The assumption is implicit that "sex labor is work" and that sexual labor is "alienated" from them, as it is from a factory worker on an hourly wage is the labor market. The labor from this perspective is not intimate but disembodied. It is impersonal and performed in the blind labor market. In other words, it is an unemotional financial transaction. Such free markets in labor are, of course, elemental to capitalist labor markets.

In an attempt to navigate around the theoretical problem of capitalism, Lazzarato (2006, p. 132) writes about unpaid "immaterial labor," which is different and "involves a series of activities that are not usually recognized as 'work'" and are missing from labor calculations. Feminist theorists, like Hochschild, in particular, have long been concerned with unpaid immaterial and affective labor, alienation by a customer who is seen as a "consumer" despite the smiles. Hochschild (2012a, 2012b) frames the issue of emotional labor as a social problem, as many works do. The question for this paper though is, "Does the Thai sex entertainment industry have similar consequences?"

What the Thai researchers described are social norms framing the type and amount of feelings appropriate for specific spaces or situations. Such "feeling rules" do emerge from a context of "emotion management" or "emotion work" (Hochschild 2012b). But emotion management is generated as a private and personal act too. Rather emotions are based on social rules regarding what is right or wrong to express in public. Through emotional labor, altering emotions, and suppressing them, people are alienated and/or estranged from the self (see, e.g., Senawong 2019; Wharton 2009). Hochschild (2012b) puts it into the language of reality construction, arguing, "when we do not feel emotion, or disclaim an emotion, we lose touch with how we actually link inner to outer reality" (p. 233). The Thai sources reviewed here consistently use the language of reality construction in a fashion that is outside the "fee for service" rhetoric of the labor market, as they create an active role in seeking intimacy.

But how deterministic is this? Do the feeling rules for Thai women for parents on the one hand and *kik* or *mia chao* arrangements adequately describe the nature of intimacy? Sharon C. Bolton (2005) questions whether capitalism took over and "appropriated all of our feelings so that there is no longer any room for sentiments, moods or reactions that have not been shaped and commodified via the 'commercialization of intimate life'" (p. 2). She further criticizes Hochschild's view of employees as she "creates an illustration of emotionally crippled actors" (p. 48) without the ability to act. However, in the case of Thai sex entertainment workers, this is not entirely the case, and we agree with Bolton's critique and with ethnographic research from inside and outside Thailand that paints a similar nuanced picture (see, e.g., Brennan 2004; Carrier-Moisan 2020; Hoang 2015; Hoefinger 2014; Muangjan 2018; Weerakulthewan 2000; Wongsuphap 1994). Hochschild's deterministic view "undervalues the vitality and independence of outlook that participants bring to organizations and neglects their ability to carve out 'spaces for resistance and misbehaviour'" (p. 62). This, again, is seen in the Thai theses by Weerakulthewan (2000), Muangjan (2018), Wongsuphap (1994) and others. It is also seen in the journalism of Wongsavun (1973), and the English language ethnography of Lapanun (2018).

Intimacy in commercial fee-for-service sexual activity assumes the focus is only on the customer and male pleasure. In personal relationships, on the other hand, both parties ideally enjoy and seek intimacy.[20,21] Examples from Thailand, where especially encounters with regular customers lead to mutual satisfaction: "I have several regular customers. It's very easy to remember faces because the place is small, and we know each other. The good thing is that I feel safe with these customers. There is sexual satisfaction too, while I make money every week" (Gkoumas 2022, p. 174). Hoefinger (2014) describes similar scenarios in the context of sex work in Cambodia with the so-called "girlfriend experience" (GFE). Bernstein (2007) describes it as a new "postindustrial paradigm" of sexual labor.

> Alongside the early modern prototype of sexual barter and the modern prototype of organized commercial prostitution, modes of commercial sexual exchange have emerged, diversified, and proliferated to create new forms of domination for many sex workers, but also (at least for some) new possibilities for creative entrepreneurship, intimacy, and community (p. 69).

> Bernstein further highlights that these new forms of commercialized sexual encounters with a high authenticity are increasingly demanded by customers. These "bounded authenticities" fulfill the client's desire for an authentic emotional and sexual relationship within a market-bound exchange (Ch. 5). As per Bernstein, there is a preference for "bounded intimate engagement over other relational forms" by the postindustrial heterosexual man. "For at least some clients, paid sex is neither a sad substitute for something that one would ideally choose to obtain in a noncommodified romantic relationship nor the inevitable outcome of a traditionalist Madonna/Whore double standard" (p. 69). These customers might even have loving and fulfilling sex with their partners at home.

In Thailand, Bernstein's "creative entrepreneurship" translates into women using affection to create intimacy with their clients, hoping for financial gain or long-term commitment simultaneously, as Weerakulthewan's (2000) research in Chiang Mai, Thailand shows. To reach this goal, women constantly negotiate the level of intimacy required by displaying emotional involvement with their customers. This might involve suppressing one's feelings to avoid tension and satisfy customers, something that is also the case in traditional boyfriend-girlfriend relationships where uncertainty and intimacy are almost inherent. Weerakulthewan (2000) writes

> On this basis, the beer bar women learn how to seek out the basic "mood of the man". They do this after they have gone through a period of adjustment in their new work. Women begin to see that there are different routes to defining relationships with different types of customers. There are at least three different categories of customers; new guest, one-time boyfriend, and sponsor. There are more mix-and-match rules, as well, and different ways to serve, such as short-time traveling, overnight trips, and holidays. The customer can propose to have sexual activity in his room, in the woman's room, short-time, schedule for another evening, etc., because this is a characteristic of working conditions in the beer bar.

> Unlike other forms of prostitution, women learn to sell their emotions. One set of relationships is reserved for western men. A woman learns to see the story in the eyes, and hopes for those men to meet the expectations of the women. Until the woman falls into her own trap, and the emotional attachment increases, and there is hope for a relationship beyond a commercial transaction (p. 190).

The relationships in sex entertainment seemingly wobble between being alienated labor, which is conducted in the marketplace, as an unsentimental cash transaction, and unalienated labor, which seeks an enduring intimate relationship.

**6. Discussion and Conclusions: Reframing the Thai Qualitative Data Regarding Intimacy in the Sexual Entertainment Industry**

Our thesis was that the "deterministic" assumptions that sexual entertainment, which assume that such work is alienated labor, or simply exploitation, is inadequate for understanding the multi-dimensional situation Thai scholars observe. In particular, such approaches miss the role intimacy plays between women, their "customers," and their larger family. This paper reframes the literature about sex work in Thailand by assuming intimacy is key to understanding why and how "sex work" emerged and is sustained. The intimacy sought in women's lives is missed whether the narrative is focused on either the "sex slavery" narrative created by activists at IJM, or the "independent entrepreneur" assumed by feminists. Notably, both types of views emphasize economic agency at the expense of other approaches. Such economically derived views are crucial, particularly for public policy advocacy. However, they also miss the very real relationships sought and sometimes found in the context of Thailand's sexual entertainment business.

Affective labor is crucial to fulfilling customers' desires and poses challenges for these women. Affective "labor" is also not by its very nature commodified. Intimacy is produced, received, perceived, and expressed differently, given pre-existing cultural meanings. It is also perceived as existing both inside and outside the labor market. Nevertheless, emotions are filtered through societal norms and our particular worldviews and belief systems. Emotions are crucial for making sense of the world and specific situations.[22]

Established research suggests that collectivistic cultures experience emotional labor and its consequences on (mental) health differently than their Western counterparts. Judging from what the Thai researchers report, this is the case here, too. Mastracci and Adams (2019) concluded in their study comparing emotional labor in East and West "that emotional labor is less stressful . . . in collectivist cultures than individualist cultures" (p. 338).

Perhaps a better way to put it is to assert that these emerging forms of work in the sex entertainment business, accelerated through capitalism, "convey an alternative perspective to the market-based 'sex work' framework, which is too often applied to situations that are sometimes more nuanced and complex" (Hoefinger 2014, p. 9).

There is still an ongoing discussion about the cross-cultural nature of work in sexual entertainment venues that mix Thai and western norms about intimacy, whether perceived as either liberating or oppressive; most intimate aspects of these women's work are still generally ignored, with few exceptions. Emerging research from the West already includes aspects of pleasure and intimacy (see, e.g., Carbonero and Garrido 2018; Kontula 2008; E. M. Smith 2017), but this topic is not yet thoroughly explored in the context of sex entertainment in Thailand. Most research instead focuses on the financial relationship between (Southeast-Asian) women and their (Western) customers, which is ultimately a reflection of the neo-colonial lens through which such questions are typically asked.

A common mistake, too, is that the typical use of "intimacy" in the context of sex work focuses on the commodified sexual act itself as the pinnacle of intimacy (Khalifian and Barry 2020; Reis and Shaver 1988). However, the actual sexual act women perform with their customers is just a fraction of the whole concept of intimacy. Much of their work is concerned with mutually beneficial "practices of intimacy" (Jamieson 2011) or, in the Thai context *dulae*. This is crucial for women in sex entertainment as they balance commodified work with their private lives outside the labor market. The "Rental Wife" quoted from *Sattahip* Rong Wonsawan is a good example. She implied that she and her friends were still unalienated not only from their sexuality but also family back home, and this was reflected in love, jealousy, desire, and intimacy outside the labor market.

And what about the intimate relationships women working in sex entertainment maintain with their parents due to the respect and filial piety so highly valued in Thailand? How do these women have intimate relationships with children, many left behind with family members? Furthermore, work and non-work friendships add another dimension to the concept of intimacy. Maintaining a good relationship with co-workers, roommates, or friends from home also involves intimacy, or as Wongsavun's informant described,

jealousy. The Thai women described in the Thai literature define their own understanding of intimacy and what it means to them during encounters with customers, in their private lives, and within themselves.

Modern work norms do not necessarily commodify the intimate aspects of the women's lives as alienated labor, as it is for a factory worker. Not even is it "emotional labor," as framed by Arlie Hochschild. In Thailand's sex entertainment business, women create relationships with their customers and extend their services with the help of affective labor seeking feelings of intimacy. The "professional girlfriends," the "rent a wife" trade during the Vietnam War in Thailand, the modern *kik,* and the committed marital relationships described by Lapanun (see Lapanun 2012, 2013, 2018) all have a place in the understanding of the Thai sexual entertainment industry. However, the fact is that customers are seemingly looking to purchase a more personalized and ultimately intimate sexual encounter. And although the intimacy is highly constructed, that imaginary bond between customer and sex worker can become real. The experiences in Thailand seem to indicate that, at times, the formerly constructed intimacy can turn into a real one.

Earning money to participate in families and return "*bun khun*" to their parents might be the driving force to enter the sex entertainment industry. However, the outlook for finding a suitable future partner through this work is a fact that should not be ignored (Knodel et al. 1997; Lapanun 2013). The construction of intimacy between the customer and the woman is part of her strategy and also the beginning of an authentic and mutually experienced venture; simultaneously, it is a commodified investment. This cultural context complicates the abuser/victim binaries of various flavors of advocates from the IJM, Empower, or the US State Department.

Thailand is still a deeply gendered society with traditional values and attributes prescribed to women (Jaisuekun and Sunanta 2021; McKenzie et al. 2021, 2022; Tangmunkongvorakul et al. 2011). As Knodel et al. (1997) highlighted, women from rural areas in Thailand see their roles as housewives, mothers, and caregivers (*dulae*), all highly respected roles. These women still see themselves as nurturers for their parents, their children, and a husband, among whom such burdens are shared, sometimes in gendered and exploitative ways, but also in the context of intimate lifelong commitments.

Western men may feel at least superficially drawn toward such nurturing attributes, believing that such values are no longer found in their Western home countries (Jaisuekun and Sunanta 2021; Lapanun 2013, 2018). However, hidden inside this are the shifts between individualistic and collectivistic countries in how these forms of labor are perceived. Western views assume that emotional labor leads to the alienation of feelings and the commodification of labor, ignoring that other cultures perceive this form of work differently (Butler et al. 2007; Mastracci and Adams 2019).

Superficially these relationships are perhaps anti-feminist, exploitative, and backward. But this perhaps also reflects a Eurocentric and neo-colonial view because it leaves out the context and location of the women involved and denies them any form of agency. The intimacy constructed between the women and their customers indirectly translates into the care (*dulae)* the women can provide for their parents' mental and physical well-being, a form of intimacy experienced, albeit on a different dimension. Being a good daughter is highly valued in Thai culture, and so is taking care of one's parents. In an economic and emotional sense, it expresses respect and appreciation. This is what Lainez (2020) called "relational work" and McKenzie et al. (2021, 2022) "relational choice". By actively providing for their families, even with the support of a customer/husband/*kik*, the status of women and their families within their home villages increases, and with it, their power (Lapanun 2012; Muangjan 2018; Spitzer 2021; Weerakulthewan 2000; Wongsuphap 1994).

Hochschild claims that emotional labor alienates workers from their inner feelings, as they play a role that ignores several aspects of her presentation of self, assuming that capitalism commodifies everything and that there is a clear line between people's outer and inner spaces (Bolton 2005; Hardt 1999; Lazzarato 2006). Implicitly, this view is shared with the anti-slavery crusaders from IJM and the Empower activists. In terms of women

working in sex entertainment, this would mean that they are clearly dividing between the work they perform, the emotions they express, and intimacy they construct, and their "real" feelings inside themselves. However, data from Thailand shows that a clear divide between performance versus actual feelings seems more blurred and nuanced.

Assuming women are alienated from their own feelings because they actively and consciously explore intimacy with their customers leaves out an important aspect of sexuality. Much research on sex work highlights the disconnection between body and mind during sexual acts during work encounters (see, e.g., Coy 2009; Ślęzak 2018; Warr and Pyett 1999). However, the Thai feminists have an alternative hypothesis, too, that women explore their sexuality with their customers. The work itself already distanced them from what is traditionally understood as a "good Thai woman". Therefore, to some extent, the work allows for liberation from the traditional understanding of how female sexuality is understood (Wongsuphap 1994; Muangjan 2018; Gkoumas 2022). Still, a go-go bar, massage parlor, karaoke club, and snooker table bar do indeed involve a job with work hours, a boss, and a paycheck. Again, we want to highlight the subjectivity of these women, who have the chance to explore how they perceive their sexuality, and inherent intimacy with one or several partners in such otherwise stigmatized contexts.

The data indicate that intimacy is crucial in Thailand's sex entertainment as it runs through the different dimensions of the work women perform like a thread and connects these. Intimacy links women with their customers, their families, and themselves. It allows for negotiating aspects of liberation, empowerment, and sexuality. And, of course, exploitation can insert itself, too—these are not mutually exclusive categories. It further creates meaning in women's daily work and opens opportunities for their future. Therefore, it is vital to include and expand on the concept of intimacy when researching sex work in Thailand. Doing so, and by not only reducing their work to the commodified sexual act, allows for a more holistic understanding of the complexity of overlapping and intersecting dimensions with intimacy at its core.

Promising research on sex work within Southeast Asia highlights well the subjectivity of women working in the field, with various studies including some aspects of intimacy within the encounters of Western customers (see, e.g., Hoefinger 2014; Law 2000; Muangjan 2018; Spitzer 2021; Weerakulthewan 2000; Weitzer 2021). These encounters have led to the spread of institutions like the go-go bar, and ultimately karaoke developed in the context of first the US military, and which emerged from more traditional forms of Thai intimacy between high-ranking Thai men and their concubines. This is a crucial finding, as women working in Thailand's sex entertainment consistently adapted working cultures to modernity's demand to reduce sex to a commercial transaction. Bernstein (2007) sees the demand for authentic (bounded) encounters like the "girlfriend experience" as a new form of sex work, emerging in modernity. Before that, she assumes sex work was mainly a commercial activity with a "monetary fee" attached. To a certain extent, we agree that modernity accelerated the demand for more personalized and commercialized sexual services. However, we doubt that these forms did not exist before, yet under different terms. In the context of Thailand, the demand for new forms of intimacy emerged with the (American) dating culture in postindustrial times as well. However, in Thailand, it is layered on the pre-modern concubinage, which shows strong characteristics of ongoing relationships with intimacy and *dulae*. Although there is a shift to more modern forms of (paid) sexual encounters, we believe that underlying forms of intimacy derived from concubinage traditions are likely still apparent in contemporary Thailand.

New forms of intimacy emerge, taking on new cultural meanings in Thailand. This is how Thai women adapt to everyday "working" experiences, seeking to maintain levels of intimacy in sex entertainment. As Muangjan (2018) highlights, Thailand's sexual entertainment business allows women to perceive themselves as "subjects" rather than "sex objects".

**Author Contributions:** Conceptualization and original drafting by P.L. Organization, review, and editing by P.L. and T.W. Theoretical overview by P.L. and T.W. Discussion and Conclusion by P.L. and T.W. Thai-English translations by T.W. All authors have read and agreed to the published version of the manuscript.

**Funding:** This research received no external grant funding.

**Institutional Review Board Statement:** No Institutional Review Board Review was required for this analytical review because no original data was collected from human subjects.

**Acknowledgments:** Petra Lemberger wishes to express her sincere thanks to Chiang Mai University/Thailand for giving her the opportunity to pursue research work during her Master's Degree Program in Women and Gender Studies at the Faculty of Social Sciences. She also wishes to thank the University for granting her the CMU Presidential Scholarship. She further wishes to thank Tony Waters for his contribution to this article, particularly the Thai-English translation of related Thai sources. Tony Waters wishes to thank Petra Lemberger for her invitation to participate in this research and the chance to visit the world of sexual entertainment as seen through her eyes. He also thanks the students and faculty at Payap University for providing a congenial place to work and study from 2016–2022.

**Conflicts of Interest:** The authors declare no conflict of interest.

## Notes

1    The Thai language has a number of dysphemisms, euphemisms, and legalisms to describe sex work/entertainment, etc. One of the official words for 'female sex workers' is *ying khai borikan* (Thai: หญิงขายบริการ), women selling service. This euphemistic term captures the variety of skills, activities, and forms of labor these women perform while embedded in the labor market.

2    Empower is a Thai sex worker organization that has been promoting rights and opportunities for sex workers since 1985. It is led and largely managed by sex workers in Thailand. The organization demands decriminalization of sex work and provides women with legal, educational, and health-related assistance, relevant to their job in the sex entertainment sector.

3    "Thailand has a greater percentage of women in senior leadership positions than both the Asia-Pacific region and the global average. In Thailand's mid-market companies, women hold 32% of senior leadership positions, higher than the global average of 27% as well as the Asia-Pacific average of 26% . . . Despite the fact that Thai women have hold executive roles in public and private sectors, they are generally still underrepresented, especially in the parliament, government, judiciary, and administration both at national and local levels. Women account for only 23.9% of high-ranking civil servants, and gender equality in senior leadership positions has risen by just 3% in the last fifteen years". (Source: UN Women Thailand https://asiapacific.unwomen.org/en/countries/thailand (accessed on 17 May 2022)).

4    "*kik*" relationships have become more common in the last 30 years or so, replacing brothel-style prostitution. The term refers to short or long-term non-exclusive relationships. Unlike in the past, both men and women can now have a "*kik*" and be a "*kik*".

5    In Thailand, massage parlors are entertainment establishments under the Entertainment Places Act 1966; in contrast to spas that offer health services.

6    Women receive a share of the amount from what clients pay for their drinks in the venue.

7    Customers usually must pay a "bar fine" to the employer or bar owner in order to allow the women to accompany them.

8    In 1959 Thailand introduced a 10-year development plan that included the tourism sector. It aimed to increase the attraction of mainly Bangkok as a tourist destination. With the increased demand for workers in the service sector, women from rural areas came to the city to serve local and international tourists in bars and other entertainment venues (Ouyyanont 2001, p. 165).

9    Among the problems with such statistics is that it assumes a discrete "sex worker" category, that exclusively occupies the work–life relative to other activities. However, any one woman may engage in many economic activities throughout the day, and especially throughout the year. Of course, the most important activity is that involves the "double burden" of providing not only cash from the labor market, but also care for family members, *dulae*, activities that are not part of the cash labor market. For the purposes of this paper, what is important is that it also excludes "grey area" sex work, which crosses over into intimate emotional arrangements. For example, when is a *mia chao* arrangement between a lonely soldier intimate and emotional, and when is it a commercial service?

10    In an attempt to quantify the broad economic impact of sex work, Empower Foundation states that approximately 80% of Thailand's sex workers are mothers and supporting five to eight other adults (Carter 2021). This impressive number was generated in the context of Empower's advocacy mission; a more nuanced interpretation would be that the Thai sex worker is contributing to a family network which includes 6–8 other adults, many of whom make financial and emotional contributions to the network, realizing their responsibilities to "care" *dulae* for each other.

11    Note: This paper specifically deals with Thai women who have civil rights, and recourse to official protection from authorities, and access to families while working in Thai sex entertainment venues. Women from neighboring countries, particularly Myanmar

do not have this recourse. They are often in Thailand on no papers, or expired papers, and subject to arrest by immigration police. They are also much less likely to speak Thai. Not surprisingly, it is among such women that the "sex slavery" practice flourishes.

12 Circumstances include rape, being a single mother who needs to support her family, but also poverty and forced labor, amongst others.

13 "In fact, with a census of nearly 60 active antitrafficking NGOs, Thailand had, as of 2007, by far the largest number of anti-trafficking NGOs of all the countries in Southeast Asia" (Padunchewit 2010, p. 6)

14 "A private, non-profit international non-governmental organization, The Asia Foundation ("TAF") was established in the year 1954 in Thailand to advance the mutual interests of the United States and countries in the Asia-Pacific region. At that time, much of its funding was routed via the CIA, with the military and political interests of the United States in mind. The Asia Foundation receives annual appropriations from the U.S. Congress and is also financially supported with contributions from corporations, foundations, individuals, and governmental organizations in the US, Canada, Europe, and Asia." (Padunchewit 2010, p. 9).

15 Although many NGOs operate within Thailand, their combating "sex slavery" results are sparse as their operation is not centralized. Most depend on sensational marketing demands to attract donor funds through churches or US government agencies. Many organizations operate independently and have Christian mission agendas (Padunchewit 2010). Furthermore, the self-identified good deeds of these NGOs should be taken with a pinch of salt because a primary point in generating data is to convince financial supporters in the United States to continue funding (see, e.g., Willis et al. 2021). This often means offering testimonials of "rescued" women on websites who attest to being victims of human trafficking. Left out is that such testimony is often in a context where testimonies help women to avoid jail terms, and are a quid pro quo for placement in NGO social programs (see, e.g., Empower Foundation 2012; Hoang 2015; Law 2000).

16 A dramatic example of idealized traditional Thai norms regarding sexuality and romance is found in the epic Thai tale *Khun Chang, Khun Phaen*, which was created in the seventeenth century, and is still important in popular Thai culture, and school curricula. The tale interweaves the tale of two high-status men, with that of a younger woman they both loved intensely and whom she loved in-return. These love affairs exist in between the battles, jealousies, infidelities, and marriages which rocked the ancient Thai Ayutthayan society. The story is one of deep commitment, love, jealousy, and intimacy.

17 Muangjan's research was conducted with women working in the "Can Do Bar" within the Empower network. The bar is used as a model under the "sex work is work" framework and advocates the decriminalization of sex work in Thailand.

18 Pseudonym.

19 Again, "*kik*" is the modern Thai word for an uncommitted but intimate sexual relationship. The word can be applied to a male or female.

20 Some research shows that having commercial sexual relationships entails a disconnection, detachment, or alienation from and dislike for the bodies—something of a reaction to the existence of intimacy rather than a denial of its significance (Coy 2009; Ślęzak 2018; Warr and Pyett 1999). This is seen as a strategy to protect the worker's mental well-being and to be able to maintain relationships outside of work. But this only makes sense in a market where labor is routinely alienated from the worker, which even Marx pointed out is a modern concept. Intimacy for some is reserved for their private lives, which they highly separate from their work life. However, even some women in more disconnected sexual scenarios report sexual pleasure during encounters with their clients, some of whom they felt attracted to (Carbonero and Garrido 2018; Ślęzak 2018; E. M. Smith 2017).

21 With the change in work structure and the demand for immaterial labor also grew new demand for a form of authentic intimacy within professional sexual encounters. One option to find this intimacy that goes beyond the simple money-for-sex transaction offer hostess services, as Carbonero and Gómez Carbonero and Garrido (2018) showed in their study in Spain. Aspects of intimate relations such as intimacy and romanticism are framed as commodities and "objects of consumption" (p. 385), mixed with the transactional nature of the cash for the commodified sex act. Nevertheless, it has to be noted that "[E]ven if prostitute's sexual pleasure was a part of commercial sex, it is not the determining criterion" (Kontula 2008, p. 611).

22 Drawing from a binary atomistic theoretical point of view with individual countable cultural traits when talking about affective labor does not work to evaluate the role of intimacy in sex work. This would seem common in a group-focused society like Thailand, where individualism is relatively low (see Hofstede et al. 2010).

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
