# Peer review of "Thailand’s Sex Entertainment: Alienated Labor and the Construction of Intimacy"

_socsci, doi:10.3390/socsci11110524_

Round 1

Reviewer 1 Report

The article entitled “Thailand's Sex Entertainment: Alienated Labor and the Construction of Intimacy” proposes to reframe the victimization/empowerment debate around women who engage in paid sexual encounters in Thailand in the light of situated socio-historical conceptions of Thai women, sexuality, and the family. The concept of intimacy in central to the author’s attempt to nuance a debate that is often based on Western feminists’ views of sex work. 

The article is very interesting and shows a deep understanding of the sociocultural and historical context of sex “work” in Thailand. I believe that the emphasis on the concepts of intimacy and emotional labor, which turns out to also be central in the argument, has the potential to make an important scholarly contribution to the fields of “sex tourism” and gender studies more broadly. 

I really enjoyed reading the paper, and believe most elements are in place to make a good contribution to the literature, but some sections need to be revised to strengthen the focus of the paper and gain more coherence overall. After reading the paper, I was still not sure what was the main objective: was it to make a theoretical contribution? Based on which materials? The research question, main argument, and the organization of the paper should be made clear from the onset, in the introduction. The introduction thus needs to be revised to better guide the reader. 

Second, until the second half of the paper, we are not sure whether this is an article based on original research, a review of extant literature or a theoretical essay. This needs to be clarified from the onset. It would also be useful to add a methodology section that explains how the review of the literature was conducted and the purpose/pertinence of the material selected for the reflection. This would be helpful to understand the research process and avoid bias. 

Another major issue with the paper is the lack of nuance in the theoretical perspectives selected to situate the reflection/argument. Only two rather radical sets of perspectives on sex tourism and sex work are presented (women as either victims or liberated). The authors are right in highlighting the shortcomings of such perspectives when applied to Thai women. However, there is too much emphasis on the « problem » of the « victim/liberated » woman dichotomy. There exists a vast literature, especially in anthropology, that goes beyond such binaries, and that draws complex pictures of the realities of women engaging in sexual activities in tourists’ areas in localities of the so called “global South”. It is thus important that the author acknowledge this literature, that highlights the nuances, complexities, tensions, as well as the interplay of emotions, love, sexuality and also sometimes exploitation, poverty, hopes and fears, etc, that arise in the everyday lives of the women in different contexts around the world (see for example the work of Brennan, Padilla, Carrier-Moisan). 

(The debate is finally situated within the context of Thailand in section 3.3.1. That is appreciated.)

Finally, the theoretical framework could be better defined and nuanced. Jamieson (2011)’S definition of the concept of intimacy is “incontournable”. And I believe the concept of emotional labor (Hochschild) might be more useful to the analysis if it was put in conversation with the work of scholars who have use it to reflect on “sex tourism” (again, Brennan’s work could be useful here). The definition of the concept is only mentioned in an end note p. 15. 

I am not altogether convinced of the relevance of Giddens concept of “plastic sexuality” here. 

On the positive side, I really like how the concept of intimacy is developed later in the paper, based on Thai worldview, instead of Western definitions of the concept. 

Some minor points:

Line 42: the West is depicted as a monolithic block. This should be nuanced.

I understand that the points raised in the introduction are developed later on in the paper, but it would be useful for the reader to see that the statements made are supported by the literature even this early in the manuscript (e.g. line 45)

Precise what is the “Wester experience” referred to line 57? 

1.1, line 87-88, There seems to be a contradiction here. In one sentence it is mentioned that sex work can be empowering for women and in the next, that women are victims. 

Line 149-150: Illustrate how modernity and women’s aspiration toward symbols of modernity are experienced in Thailand in terms of desires, hopes, subjectivity, etc. 

Other reference of interest: The introduction to the book “Modern loves”, by Hirsh and Wardlow.

Line 470 : see Carrier-Moisan for a similar analysis of sexual encounters in tourists areas in Brazil. 

Line .550 : from this point the author starts citing other research at length. Because there is not methodology section, it comes as a surprise, and we do not fully understand why. 

Line 810-820 : this long quotation is already cited. There is no need to repeat it. 

Thank you for this important work. I look forward to reading the revised version. 

Author Response

Dear Reviewer,

Thank you very much for your helpful feedback and encouragement in regards to our analytical essay. We highly appreciated your comments and have incorporated revisions as follows below.

Kindly note that all changes are highlighted in yellow– some paragraphs were moved during re-organization, whereas some are entirely new.

The article entitled “Thailand's Sex Entertainment: Alienated Labor and the Construction of Intimacy” proposes to reframe the victimization/empowerment debate around women who engage in paid sexual encounters in Thailand in the light of situated socio-historical conceptions of Thai women, sexuality, and the family. The concept of intimacy in central to the author’s attempt to nuance a debate that is often based on Western feminists’ views of sex work.

 The article is very interesting and shows a deep understanding of the sociocultural and historical context of sex “work” in Thailand. I believe that the emphasis on the concepts of intimacy and emotional labor, which turns out to also be central in the argument, has the potential to make an important scholarly contribution to the fields of “sex tourism” and gender studies more broadly.

 I really enjoyed reading the paper, and believe most elements are in place to make a good contribution to the literature, but some sections need to be revised to strengthen the focus of the paper and gain more coherence overall. After reading the paper, I was still not sure what was the main objective: was it to make a theoretical contribution? Based on which materials? The research question, main argument, and the organization of the paper should be made clear from the onset, in the introduction. The introduction thus needs to be revised to better guide the reader.

We have re-organized our analytical essay to be more coherent and more explicit in our approach. This includes:

  • Revision of Introduction
  • Specifying a formal Statement of the Problem
  • Adding a paragraph on Research Aim and Methodological Approach
  • Specifying Theoretical Framework.
  • Re-organizing several sections

Second, until the second half of the paper, we are not sure whether this is an article based on original research, a review of extant literature or a theoretical essay. This needs to be clarified from the onset. It would also be useful to add a methodology section that explains how the review of the literature was conducted and the purpose/pertinence of the material selected for the reflection. This would be helpful to understand the research process and avoid bias.

  • As mentioned above, we added our Research Aim and Methodological Approach

 Another major issue with the paper is the lack of nuance in the theoretical perspectives selected to situate the reflection/argument. Only two rather radical sets of perspectives on sex tourism and sex work are presented (women as either victims or liberated). The authors are right in highlighting the shortcomings of such perspectives when applied to Thai women. However, there is too much emphasis on the « problem » of the « victim/liberated » woman dichotomy. There exists a vast literature, especially in anthropology, that goes beyond such binaries, and that draws complex pictures of the realities of women engaging in sexual activities in tourists’ areas in localities of the so called “global South”. It is thus important that the author acknowledge this literature, that highlights the nuances, complexities, tensions, as well as the interplay of emotions, love, sexuality and also sometimes exploitation, poverty, hopes and fears, etc, that arise in the everyday lives of the women in different contexts around the world (see for example the work of Brennan, Padilla, Carrier-Moisan).

We have included recommended anthropological works to give a more nuanced picture of sex work with examples from Latin America and the Caribbean. However, we did not elaborate on these studies at length, as our focus is Southeast Asia, particularly Thailand. Therefore, we also included works from SEA. But, as is our point, there are not many similar studies from Thailand, as they are mainly within the “victim/liberated” woman dichotomy; apart from the original Thai sources, which we discuss in length.

(The debate is finally situated within the context of Thailand in section 3.3.1. That is appreciated.)

Finally, the theoretical framework could be better defined and nuanced. Jamieson (2011)’S definition of the concept of intimacy is “incontournable”. And I believe the concept of emotional labor (Hochschild) might be more useful to the analysis if it was put in conversation with the work of scholars who have use it to reflect on “sex tourism” (again, Brennan’s work could be useful here). The definition of the concept is only mentioned in an end note p. 15.

Thank you for this recommendation. We included Jamieson’s intimacy concept and reflected more on Hochschild’s work.

I am not altogether convinced of the relevance of Giddens concept of “plastic sexuality” here.

We removed this part, as explaining the relation of Gidden’s concept in connection with the article would have been too lengthy.

 On the positive side, I really like how the concept of intimacy is developed later in the paper, based on Thai worldview, instead of Western definitions of the concept.

 Some minor points: Line 42: the West is depicted as a monolithic block. This should be nuanced.

Precise what is the “Western experience” referred to line 57?

Where the term “Western” is unavoidable, we have focused “Western individualism” as defined by Hofstede et al.  In other places, we have taken out the term “Western,” or replaced it where appropriate with American.  We agree that the term “Western” is usually too general.

 I understand that the points raised in the introduction are developed later on in the paper, but it would be useful for the reader to see that the statements made are supported by the literature even this early in the manuscript (e.g. line 45)

Introduction is updated to guide the reader into the discussion.

 1.1, line 87-88, There seems to be a contradiction here. In one sentence it is mentioned that sex work can be empowering for women and in the next, that women are victims.

This paragraph was removed during the re-organization

Line 149-150: Illustrate how modernity and women’s aspiration toward symbols of modernity are experienced in Thailand in terms of desires, hopes, subjectivity, etc.

We moved this section to the Statement of the Problem. There we emphasized on intimacy and relationships without going too deep into other aspects of modernity.  The new reference to Hofstede et al (2010) with respect to individualism are also relevant.

Other reference of interest: The introduction to the book “Modern loves”, by Hirsh and Wardlow.

Thank you for this recommendation. We read the reviews online and agree that it is relevant.  However, due to time/availability constraints, we could not include this source in the article, but we noted it for further reference and future research.

Line 470 : see Carrier-Moisan for a similar analysis of sexual encounters in tourists areas in Brazil.

This is a provocative source, but we could not find a similar analysis in this source and don’t quite see how it would fit.

Line .550 : from this point the author starts citing other research at length. Because there is not methodology section, it comes as a surprise, and we do not fully understand why.

Clarified in the methodology section

 Line 810-820 : this long quotation is already cited. There is no need to repeat it.

Removed

Thank you for this important work. I look forward to reading the revised version.

Reviewer 2 Report

The authors try to provide a more holistic and complex point of view about the motivations and behaviors of women in the sex industry in Thailand. This vision moves away from the traditional Western perspective about prostitution, more adapted to the culture of these women.

The article presents an original and interesting perspective on the subject that leads us to reconsider our vision of prostitution in Thailand. I recommend reviewing the text to correct minor errors (punctuation, spaces...)

Author Response

Dear Reviewer, 

Thank you for your positive feedback. We have updated minor errors as recommended. 

Round 2

Reviewer 1 Report

The author has done excellent work restructuring their paper as to respond to the issues raised in my first report. The main objective of the paper is now much clearer, the concepts better defined and the argument more nuanced. 

Author Response

Thank you for the positive response.